# Sharing Is Caring: Exploring Distributed Solar Photovoltaics and Local Electricity Consumption through a Renewable Energy Community

Evandro Ferreira *[ID], Miguel Macias Sequeira [ID] and João Pedro Gouveia [ID]

CENSE—Center for Environmental and Sustainability Research & CHANGE—Global Change and Sustainability Institute, NOVA School of Science and Technology, NOVA University Lisbon, Campus de Caparica, 2829-516 Caparica, Portugal; m.sequeira@campus.fct.unl.pt (M.M.S.); jplg@fct.unl.pt (J.P.G.)
* Correspondence: ec.ferreira@campus.fct.unl.pt

**Abstract:** Renewable Energy Communities (REC) can play a crucial role in enhancing citizen participation in the energy transition. Current European Union legislation enshrines energy communities and mandates Member States to encourage these organizations, promoting adequate conditions for their establishment. Nevertheless, uptake has been slow, and more research is needed to optimize the associated energy sharing. Using a Portuguese case study (REC Telheiras, Lisbon), this research aims to match local generation through four photovoltaic systems (totalizing 156.5 kWp of installed capacity) with household electricity consumption while cross evaluating the Portuguese legislation for energy sharing. The latter aim compares two scenarios: (a) current legislation (generated energy must be locally self-consumed before shared) and (b) equal share for members with a fixed coefficient. The evaluation is performed according to two indexes of self-consumption (SCI) and self-sufficiency (SSI), related to the simulation of four photovoltaic systems in public buildings, their associated consumption profiles, and an average household consumption profile of community members. The results show that, while maximizing self-consumption for the same values of generation and consumption, the number of participants is considerably lower for Scenario A (SCI = 100% is achieved with at least 491 residential members in Scenario A and 583 in Scenario B), implying that legislative changes enabling energy communities to better tailor sharing schemes may be necessary for them to become more attractive. The methods and results of this research can also be applied to other types of facilities, e.g., industrial and commercial consumers, if they are members of a REC and have smart meters in their installations.

**Keywords:** renewable energy community; prosumers; collective self-consumption; energy sharing; Portugal

## 1. Introduction

Climate change and sustainable development are two of this century's most critical challenges [1,2]. In this context, the urgency of a deep energy transition is increasingly evident [3]. This process should be based on economic, social, and environmental pillars, involve all citizens, and promote social change through enhancing energy democratization and mitigating energy poverty [4].

Distributed generation plays a vital role in this transition, primarily through renewable energy sources by the so-called prosumers (electricity consumers who also start to generate energy) [5]. These microgeneration systems have important aspects, such as (a) reduction in greenhouse gas emissions because most of these systems are based on renewables, also enhancing the share of renewable sources in the matrix [6]; (b) promoting lower ohmic losses in the distribution grid, since electrical resistance is proportional to the length of the conducting wire, reducing the distance between the generation site and the consumption point reduces losses in energy distribution [7,8]; and (c) providing higher energy security

and affordability, due to the vast number of microgeneration sites in opposition of fewer large generation plants (technical and operational faults may be a significant problem for centralized energy generation) [9].

Consistently lower equipment prices, a high level of modularity, a relatively easy installation, and elevated social acceptance are important characteristics of solar photovoltaic (PV) systems that explain the broad application of this renewable source in distributed generation systems [10,11]. Nevertheless, a key challenge of PVs applied to residential prosumers is the observed mismatch between the generated energy profile and the energy demand profile: PV generation starts at the beginning of the morning, facing a peak close to midday (higher levels of irradiance) and ending with dusk [12]. In contrast, the average energy household demand profile presents one peak in the early morning (citizens using electricity for morning routines) and another one during the evening (citizens coming back home) [13]. In the context of an increasing number of distributed energy systems based on PVs, this mismatch may lead to problems associated with reverse power flows in the low voltage grid in neighborhoods with a high concentration of prosumers due to high amounts of energy injected into the grid during the PV peak hours. This can lead to negative impacts in the aging process of distribution transformers and highlights the importance of an adequate PV system sizing according to the energy balance between consumption and generation [14,15].

Another vital topic for distributed generation systems is the form of compensating the surplus energy (excess generation in a self-consumption structure), where the so-called net metering and net billing policies are the most common schemes [16]. In net metering, the surplus energy in a specific billing period is translated into "energy credits" that can be consumed in a particular time (hours, months, or even years); basically, the generated energy has the same value, whether self-consumed or injected into the grid [17]. However, in the net billing scheme, the surplus energy can be sold at a tariff that can be fixed or variable and usually lower than the consumption tariff from the grid [18]. It is important to mention that both schemes vary according to the national energy sector policies. For example, in Brazil, the net metering scheme allows the consumption of "energy credits" for five years [19]; in Portugal, the match between consumption and generation is based on 15 min, with the surplus energy being directed to a net billing scheme where it can be injected into the grid without any payment to the prosumer, sold directly through bilateral contracts or sold through specific vendors in the electricity market at a fixed tariff (considerably lower than the consumption tariff) or at a tariff that varies with the fixed market price according to the injection time (based on electricity supply and demand for the Portuguese context in that day and hour), but still lower than the consumption tariff [20].

Due to these differences, the overgeneration of electricity through a distributed generation system is less valuable to a prosumer in a net billing scheme than in a net metering scenario every time the prosumer does not sell the surplus (free injection into the grid) or the surplus sale tariff is lower than the one for energy consumption [18]. This aspect highlights the importance of maximizing both self-consumption and self-sufficiency of the prosumer installation, which can be measured through the Self-Consumption Index (SCI)—the ratio between the self-consumed parcel of the locally generated energy and the total local generation—and the Self-Sufficiency Index (SSI)—the ratio between the self-consumed parcel of the locally generated energy and the total energy consumption of the installation [21].

A pillar for the success of the current energy transition is the enhancement of energy democratization, where citizen's active participation is crucial for translating the population's real needs and making the process fairer and more inclusive [22,23]. In this context, collective energy actions at the local scale, such as renewable energy communities, are essential to bring together citizens in favor of higher participation and more active roles in energy decisions [24]. In the European Union (EU), a Renewable Energy Community (REC) is briefly defined as a legal entity with open and voluntary participation that is controlled by its members in an autonomous and effective manner [25]. Its primary goal

must be to provide environmental, social, and economic benefits to its members—who can be households, small and medium enterprises, and local governments—instead of aiming for financial profits [26]. These organizations can act in different areas, such as generating and sharing renewable energy, promoting energy efficiency practices, enhancing the energy literacy of the members, applying sustainable transport solutions locally, developing smart grid concepts, and implementing energy storage technologies [27]. Nevertheless, research on the actual operation of real-world RECs is still scarce, particularly regarding the technical and economic optimization of electricity generation and sharing according to specific national contexts.

RECs can be associated with important benefits to the community and its members in different dimensions, namely economic (e.g., savings on energy bills, profits from the renewable generation system after reaching the payback time of the system, possibility to sell the surplus energy to the grid), social (e.g., partnerships with local authorities to provide local development, energy poverty mitigation, social inclusion), and environmental aspects (e.g., reduction in carbon emissions, renewables acceptance and environmental consciousness between the members) [28].

There is also an important link between renewable energy communities and energy poverty mitigation: since RECs involve collective energy generation and sharing based on renewable energy sources, they also provide a more reliable and affordable supply of electricity to its members while turning citizens into active players in the transition and enhancing their energy literacy. RECs can act to alleviate energy poverty locally, such as Enercoop in France (microdonations from consumers and producers to associations that act to mitigate energy poverty locally), ZEZ in Croatia (training courses for the unemployed to become energy advisors to low-income households), and Som Energy in Spain (works together with local authorities to identify and help families in energy vulnerability situation) [29].

Being a global leader in the implementation of RECs, the EU needs to have different approaches to the reality of each Member State to analyze the roll-out of these energy communities, where each EU Member, according to the respective characteristics and national regulation, aims to support and foster the creation of new RECs inside its territory [30]. For example, northern countries such as Germany, Denmark, the Netherlands, and Sweden have the most significant number of citizen-led RECs due to their cultural traditions of collective participation and citizen engagement. However, even though higher solar irradiation levels are found in southern countries such as Portugal and Spain, a smaller number of RECs are operating in these Member States [31].

According to the Joint Research Centre of the European Commission, there were about 3500 renewable energy cooperatives in Europe in 2020. Many other structural forms of RECs also are found in the EU, such as eco-villages, small-scale heating organizations, and small-scale renewable energy projects based on energy sharing [31]. The European Union itself also has a crucial role in spreading these energy organizations through their developing and funding projects associated with RECs and other citizen-led initiatives related to renewable energy systems [32]. All of these pioneering actions serve as a basis for creating similar projects outside the EU, paving the way for worldwide growth in the number of RECs [31].

In Portugal, the European legislation was transposed to national law in 2019 and 2022, allowing for the development of energy community projects in the country [33,34]. Nevertheless, this transposition has been criticized for its substantial deficiencies, including a lack of clarity, the risk of overtaking by market actors, neglecting the principle of autonomy, and strict geographical proximity criteria, among other aspects [35]. Recently, a few projects have already been operational, with several challenges constraining their development [28]. It is important that different Member States exchange ideas regarding the REC's national legislation, aiming to replicate the strong points while adapting to each context [36]. In Italy, for example, regional energy authorities can also create specific laws to promote the creation of RECs in their territory, as in the example of the Piedmont and Puglia regions [37]. The

French case plays a pioneering role in collective energy organizations; in 2015, incentives to be part of local renewable energy projects were already established [38]. In Germany, a well-developed scheme of support for developing citizen-led energy organizations is highlighted by its national legislation regarding RECs, resulting in standardized and considerably easier implementation and licensing processes. Different scenarios also occur inside the EU; in Bulgaria, in 2021, there were still no specific national policies to support the development of RECs [39].

Furthermore, for collective self-consumption schemes, the current Portuguese legislation implies that all generated energy must be self-consumed in the generation site before being shared with other participants, based on a 15 min analysis [40]. We hypothesize that this can be a significant barrier for RECs, mainly when the PV systems are in buildings with high energy consumption levels. During the day, there may be extremely low amounts or even zero surplus energy to share with the members, resulting in decreased SCI and SSI indexes and more extended payback periods for the members.

A real case study of an emerging REC in Portugal is the Telheiras Renewable Energy Community (REC Telheiras, Lisbon), which focuses on sharing PV-generated electricity between local consumers [41]. Since REC Telheiras functions with collective investment by its members for the acquisition and installation of PV systems, in addition to annual maintenance and operation costs, evaluating the REC's overall SCI and SSI according to the member's consumption profiles is essential to guarantee an attractive payback time, thus enabling new investments for future expansions and participation of new members.

In this context, this research models the potential numbers of residential members participating in REC Telheiras under two scenarios: (a) following the current Portuguese legislation for energy sharing inside a REC and (b) sharing the PV generation through an equal and fixed coefficient for all the members. It aims to provide an analysis of SCI and SSI indexes for both scenarios and different numbers of members according to local generation capacity and consumption patterns. Scarce research has examined the matching between PV energy generation and electricity consumption for the case of RECs, mainly using detailed 15 min consumption data from its members and comparing different scenarios according to national legislation.

Similar studies have been conducted regarding self-consumption and self-sufficiency indexes evaluation in communities based on renewable energy sharing, as presented in [42] for a household's energy sharing inside a Portuguese REC, focusing on the interplays between prosumers and economic analysis, in [43] for the evaluation of SCI and SSI based on PV systems with storage in a REC in Italy, and in [44] for analyzing SCI and SSI for RECs constituted by members that live in the same building and have a PV installation in the common roof area for different regions of Spain. Other studies that analyze SCI and SSI in other scenarios were produced, such as an evaluation of a PV system sizing according to those indexes for a subway station in Romania [45] and a study about the impact of SCI and SSI related to the installation of an electric vehicle charging station for a household PV prosumer [46].

This paper is structured as follows. Section 2 presents the case study, materials, and methods used, namely to size PV generation, analyze consumption profiles, analyze the number of members according to the SCI and SSI indexes, and compare two scenarios, also providing a financial comparative analysis. Section 3 presents and discusses the multiple results obtained. Finally, Section 4 ends with conclusions and recommendations for further analysis regarding RECs.

## 2. Materials and Methods

### 2.1. Study Case: REC Telheiras

The Telheiras neighborhood is located in the north of Lisbon in one of its most populous civil parishes (Lumiar Civil Parish). Its urbanization was planned in the 1970s by the Public Urbanization Company of Lisbon, where most of the buildings date from the 1980s to 1990s and suffer from poor construction characteristics and thermal discomfort

both in summer and winter. With around 50,000 inhabitants, the civil parish's households are heterogeneous, ranging mostly from low to middle classes, with around 21% of its population over 65 years old and only 5% being migrants [47].

Created by a volunteer group of local citizens with the initial idea of "let's produce our renewable energy locally and share among neighbors", the Telheiras Renewable Energy Community's primary purpose is to contribute to a fairer energy transition and mitigate energy poverty locally through PV renewable energy sharing between its members. The idea emerged in 2020 during a local event in the Telheiras neighborhood to collect ideas from citizens about new projects and activities to be developed in the area [48].

Currently, the REC Telheiras pilot project is in the final part of the licensing phase with the responsible national authorities [49]. This pilot project includes 17 members: the public building itself and 16 local families, three of whom are energy-poor. The aim was for the generated energy to be shared equally among all members; however, due to the current Portuguese legislation, the public building must use the generated energy first, with only the surplus shared with the remaining members.

Important factors of the development of REC Telheiras are (a) collaboration between the Local Partnership of Telheiras (a network of local non-profit organizations), local authorities, namely the Lumiar Civil Parish, framed within a technical assistance process of the EU Energy Poverty Advisory Hub (EPAH) and supported by a diversity of multi-thematic experts (engineering, regulation, and financing) [50]; (b) an already existing high level of engagement in the community regarding practices associated with sustainability, environmental awareness, and circular economy, which helped to create a volunteer group of locals to start the development process of the REC; and (c) active leadership of the coordination group, crucial to maintain the engagement of the volunteers and progress in a constant and proactive manner in solving challenges associated with the implementation process.

As set during its early development stages, during the installation phase of the REC, all members invest a certain amount associated with a fixed sharing coefficient related to the expected PV-generated electricity. Since one of the main objectives of REC Telheiras is helping to mitigate energy poverty at a local scale, social members (families in a context of energy vulnerability) are also part of the organization, where the costs associated with their participation (i.e., investment) are shared between the regular members and the local civil parish, which is also a member of the REC and user of the building. Then, during the operational phase, all members receive a parcel of the generated energy and pay an annual quota for maintenance, insurance, and grid access fees, where social members pay a reduced amount. All the costs associated with equipment acquisition and installation will be paid through members' investment, while maintenance and operational costs are associated with an annual quota also paid by the members [49].

### 2.2. PV Systems Sizing and Simulation

Four buildings intended to install solar PV systems were selected according to the legal viability of utilizing the roof area, suitable conditions for PV generation, and its location inside the neighborhood. The chosen buildings were a community center (a building that serves as a venue for local events and social activities), a local government headquarters (local administrative center), a gymnasium (local multi-sports pavilion for sports practices), and a local primary school, where these buildings will be denoted as Building A, B, C, and D, respectively. Figure 1 presents views of the four selected public buildings. The pilot project is currently in the licensing stage and corresponds to Building A.

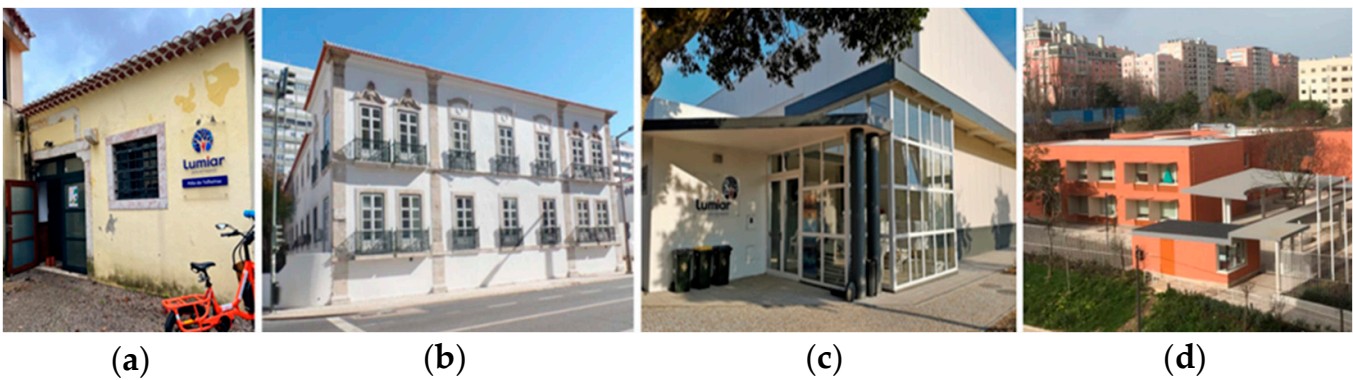

**Figure 1.** Views of the selected public buildings for PV generation: (**a**) community center; (**b**) local government headquarters; (**c**) gymnasium; (**d**) local primary school.

With the aim of selecting the most suitable PV module for the four systems, different solar cells technologies were analyzed. The various types of solar cells are divided into four generations according to their technological evolution [51]. First-generation solar cells represent more than 80% of the actual PV market, divided into two types: polycrystalline and monocrystalline silicon cells, where the latter have higher nominal generation efficiency [52]. Solar cells of the second generation, such as amorphous silicon cells, copper indium gallium selenide cells and cadmium telluride ones, are based on thin films, with associated lower efficiencies than the ones of the first generation and a smaller and decreasing presence in the PV market [53]. Although third and fourth generation solar cells (organic cells, perovskites, and quantum dot-based cells) have a promising trend for the future years, they still have an almost negligible availability in the actual market [51]. Therefore, in the current market conditions, the most suitable solar cell technology for the PV systems of REC Telheiras is a monocrystalline module.

The selected PV module for all four systems was the Risen RSM-150-8-500-M (monocrystalline silicon) since it has a high nominal efficiency (20.3% at Standard Test Conditions (STC)) and a relatively high nominal power (500Wp at STC), in addition to an interesting cost–benefit relation and wide availability in the Portuguese solar market. Parameters and details of the selected module are presented in Table 1 [54].

**Table 1.** Parameters of the selected PV modules for REC Telheiras [54].

| PV Module: Risen RSM 150-8-500-M (STC Conditions) | | |
|---|---|---|
| Nominal power: 500 Wp | Open circuit voltage: 51 V | Monocrystalline cells |
| Efficiency: 20.3% | Short circuit current: 12.5 A | Area: 2.49 m$^2$ |

The sizing and simulation processes were performed through a specific photovoltaic software (PVSyst version 7.4) [55] and according to four main criteria: available roof area, shading conditions, orientation of the roof, and limitations of the building (constructive aspects and electrical installation conditions), aiming to maximize the PV installed capacity in each building. The characteristics of each building according to the selected criteria are presented as follows:

- Building A—Community Center: Although this building has a considerable roof area, a significant portion of it requires repairs. Therefore, only a small southern portion of the roof is available for the PV system, where the only shading element is the chimney of the building. This roof area has two sides (one is oriented south, and the other is approximately rotated 20° northeast), and the structural and electrical conditions of the building are well-preserved.
- Building B—Local Government Headquarters: a considerable roof area is available, with significant shading incidence in some portions due to higher buildings in the

surroundings. Thereby, PV modules should be installed in the southern and northern areas, respectively oriented to north and south. This building also has adequate structural and electrical conditions.

- Building C—Gymnasium: a very significant two-sided roof area is available (oriented to northeast and southwest), without shading elements in the roof or the surroundings and presenting good conditions of the structure of the building and the electrical installation. However, possible roof reinforcements might be required since the large available roof area will produce a relatively higher value of PV installed capacity.
- Building D—Local Primary School: many different roof portions with different sizes and orientations are available for PV modules. Here, the different height levels of adjacent roof areas and three specific trees located inside the school promote areas with intensive shading effect, where two of the selected roof areas are slightly oriented south, and the other two are entirely flat. The constructive aspects and the electrical installation of this building also are in a generally good condition.

In addition, a financial analysis of the simulated systems was also performed, through consulting the unit prices of equipment and parts required for the respective PV systems. A summary of these costs can be seen in Table 2.

**Table 2.** Unitary prices for the components of the PV systems.

| Components | EUR/Unit | Reference |
|---|---|---|
| PV Module Risen RSM 150-8-500-M | 240.00 | [56] |
| Inverter Sungrow SG8.0 RT | 1142.58 | [57] |
| Inverter Sungrow SG20.0 RT | 1628.24 | [58] |
| Inverter Growatt 70KTL3-X | 4797.00 | [59] |
| Inverter Sungrow SG25.0 RT | 1991.36 | [60] |
| Aluminium Profile 2.08 m | 11.50 | [61] |
| End Clamp 30/35 mm | 0.97 | [62] |
| Middle Clamp 30/35 mm | 0.51 | [63] |
| Roof Hook Stainless Steel | 4.52 | [64] |
| Solar Cable 6 mm$^2$ Black (m) | 1.71 | [65] |
| Solar Cable 6 mm$^2$ Red (m) | 1.71 | [66] |
| Protection Cable 6 mm$^2$ Yellow/Green (m) | 1.58 | [67] |

It is important to mention that the installation, maintenance, and operation costs are not included in the values presented in Table 2. For the Portuguese context, installation costs can be estimated as 5% of the total cost of the equipment [68].

The energy compensation analysis of the REC will be carried out separately. Thus, all four simulations were performed without considering self-consumption; only values of the generated energy in each building were obtained. The utilized software generates results for each month and yearly averages but also provides the average daily generated energy hour by hour of each month, as well as the Performance Ratio (PR) of the systems (ratio between the actual energy generation after all conversions processes and the theoretically possible energy output). In addition, 3D images of the respective roofs with the PV modules were obtained by combining the results obtained in PVSyst with images reserved from Google Earth.

Since the utilized software for sizing and simulation of the systems gives hourly average values for an entire day for each month and the consumption measurement period is 15 min, each hourly measurement of generated energy was divided into four equal 15 min measurements corresponding to a quarter of the hourly value obtained through the simulation.

*2.3. Consumption Profiles: Participating Households and Public Buildings*

In order to analyze electricity consumption in REC Telheiras, the consumption profiles of 8 of the 13 household members of REC Telheiras' pilot project was obtained through the online portal of the demand side operator, which contains data for the respective smart meters with all the 15 min measurements for a year. The analyzed consumption cycle was from January 2023 to December 2023. The consumption registries were retrieved at the end of January 2024 from the Distribution System Operator (DSO) platform. Following the same approach, the consumption profiles of all four selected public buildings were also extracted.

Two critical assumptions were considered for this consumption analysis: (a) the average profile of the available eight members was considered as the average household consumption profile for one typical member of the REC, and (b) a representative consumption day for a single member and each analyzed public building was defined for each month through average values of every 15 min periods associated with all the days of the respective month.

*2.4. Energy Compensation: SCI and SSI Indexes*

According to the following description, the energy compensation (generation minus consumption) was analyzed for each scenario. In "Scenario A: Current Portuguese legislation for energy sharing" the following approach was defined:

1. Obtain the energy compensation for each 15 min for each public building and analyze the respective surplus energy.
2. Calculate the SCI and SSI indexes for each 15 min period for each public building and average daily and monthly values.
3. Define an arbitrary initial number of members in the REC and the associated fixed and equal coefficient share for members of the total surplus energy.
4. Obtain the energy compensation evaluation for each 15 min considering the total surplus energy and the total energy consumption of the considered number of members.
5. Calculate the SCI and SSI indexes for each 15 min period for that specific number of members and average daily and monthly values.
6. Vary the number of members aiming to maximize the monthly and yearly SCI and SSI indexes for the REC.

In "Scenario B: Share between members and public buildings with a fixed coefficient", we explore the implications of a hypothetical scenario where the Portuguese legislation would allow for equal energy sharing for all REC members, including the generation sites themselves. The approach taken was:

1. Define an arbitrary initial number of members in the REC and the associated fixed and equal coefficient share for the total energy consumption.
2. Obtain the energy compensation for each 15 min period considering the total energy generation and the total energy consumption of the number of members considered.
3. Vary the number of members, aiming to maximize the monthly average SCI and SSI indexes for the members.

This evaluation for each scenario was completed for ten different numbers of REC members, aiming to provide an overview of the REC size possibilities while varying the values for SCI and SSI. For both scenarios, SCI and SSI were calculated through Equations (1) and (2), respectively.

$$\text{SCI (\%)} = \text{Self-consumed Energy} / \text{Total Generation} \qquad (1)$$

$$\text{SSI (\%)} = \text{Self-consumed Energy} / \text{Total Consumption} \qquad (2)$$

### 2.5. Scenarios Comparison

Scenario A and Scenario B were compared according to the obtained values for SCI and SSI for the analyzed number of members in the REC. In Portugal, the net billing scheme usually has a considerably lower tariff for sold surplus energy; the main objective is to obtain high SCI values, meaning that most of the generated energy is being self-consumed or completely avoided injection into the grid (SCI = 100%).

A comparative cost–benefit analysis between both scenarios was also performed. The payback times for the participating families and for the local government were calculated according to the respective investment costs and annual costs, and to the electricity bills savings associated with the consumption of electricity from the REC and considering the Portuguese-regulated market tariff of the first semester of 2024 (0.1625 EUR/kWh [69]). It is also important to account that families need to pay a grid access fee when participating in a REC (in the first semester of 2024, this fee was 0.0106 EUR/kWh [69]; the local government does not pay grid access tariffs because it is producing and consuming electricity in the same building and thus not using the grid. In Scenario A, the investment paid by the local government for each public building and the investment paid by the families are associated with the respective percentages of the renewable generation that is consumed, while in Scenario B it is divided equally between families and public buildings. Annual maintenance and operational costs in the Portuguese context are estimated at around 3% of the total investment per year and are also part of this financial analysis [68].

## 3. Results and Discussion

### 3.1. PV Systems Sizing Results

The four PV systems were sized and simulated independently and are denoted as System A (Building A), System B (Building B), System C (Building C) and System D (Building D). System A was chosen as part of the REC's pilot project.

### 3.1.1. PV System A—Community Center

System A was sized and simulated according to the previously presented building characteristics, resulting in 8.0 kWp of nominal installed peak power (16 modules) and an expected annual generation of 12,126 kWh, with an average yearly PR of 83.5%. Table 3 presents the results of the PV simulation for System A.

**Table 3.** PV simulation results—System A (monthly electricity generation).

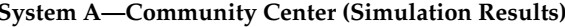

**System A—Community Center (Simulation Results)**

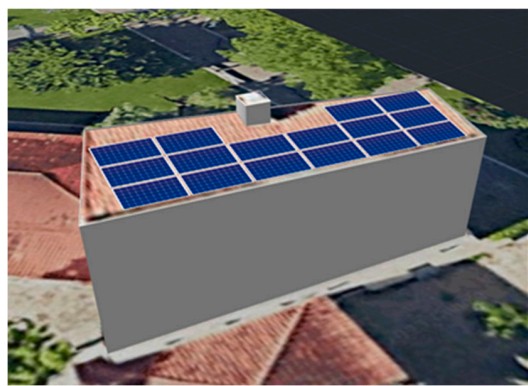
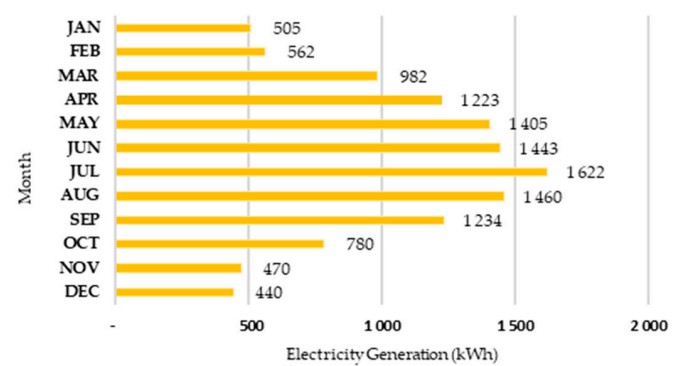

### 3.1.2. PV System B—Local Government Headquarters

After this analysis, System B was sized and simulated, resulting in 24.0 kWp of nominal installed peak power (48 modules) and an expected annual generation of 37,636 kWh, with an average yearly PR of 84.7%. The obtained results are shown in Table 4.

**Table 4.** PV simulation results—System B (monthly electricity generation).

**System B—Local Government Headquarters (Simulation Results)**

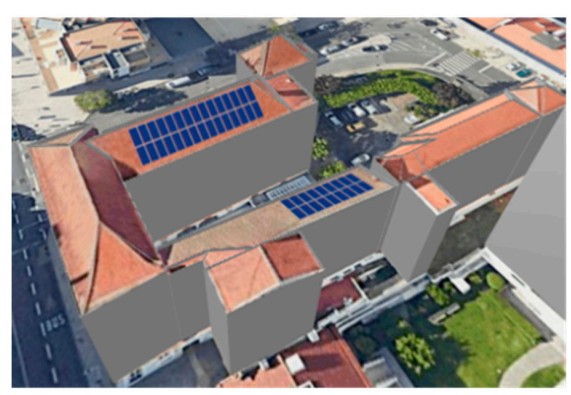

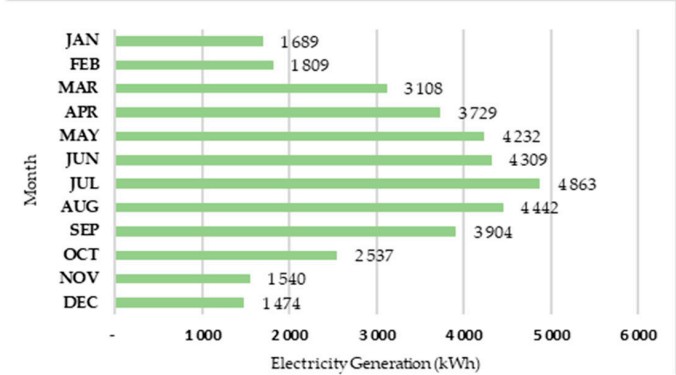

### 3.1.3. PV System C—Gymnasium

Thereby, System C was sized and simulated, resulting in 78.0 kWp of nominal installed peak power (156 modules) and an expected annual generation of 109,263 kWh, with an average yearly PR of 84.4%. Table 5 shows the obtained results for System C.

**Table 5.** PV simulation results—System C (monthly electricity generation).

**System C—Gymnasium (Simulation Results)**

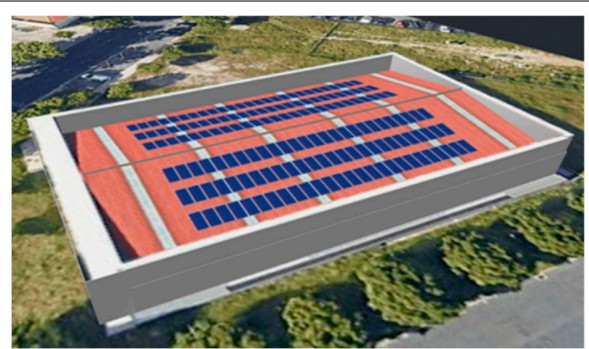

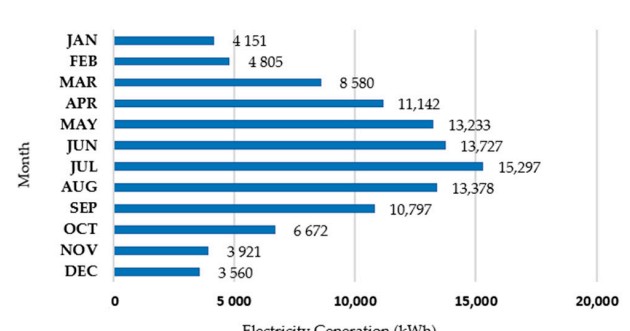

### 3.1.4. PV System D—Local Primary School

Finally, System D was sized and simulated, resulting in 46.5 kWp of nominal installed peak power (93 modules) and an expected annual generation of 68,399 kWh, with an average yearly PR of 84.7%. Table 6 presents the results of the simulation of System D.

### 3.1.5. Overview and Total Electricity Generation

All four systems' expected generation is 227,424 kWh/year, corresponding to a total installed capacity of 156.5 kWp and 313 solar PV modules. As aforementioned, the total monthly values of PV generation are shown in Figure 2, and the average generation values for the 15 min daily measurements for a typical day of three months (January, March, and December) are presented in Figure 3.

**Table 6.** PV simulation results—System D (monthly energy generation).

**System D—Local Primary School (Simulation Results).**

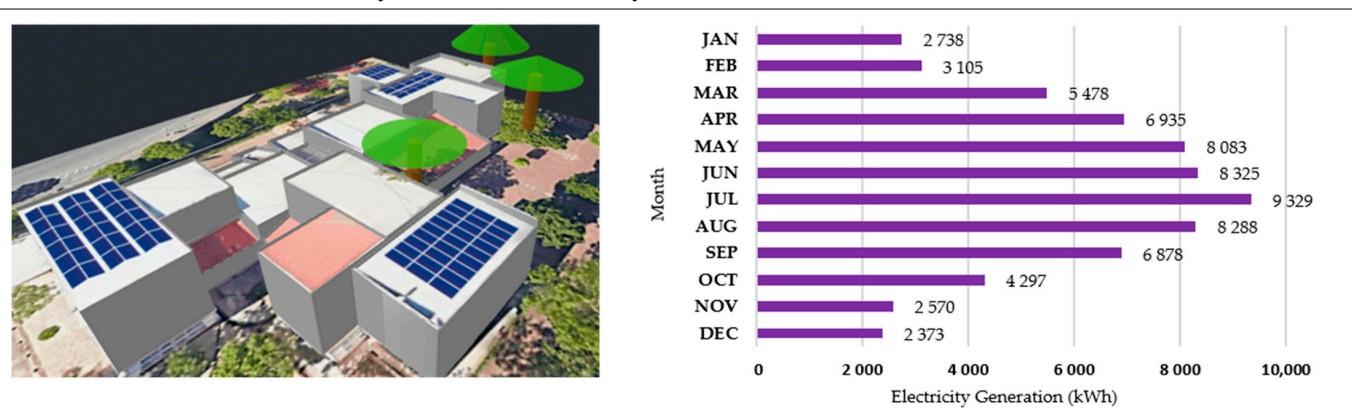

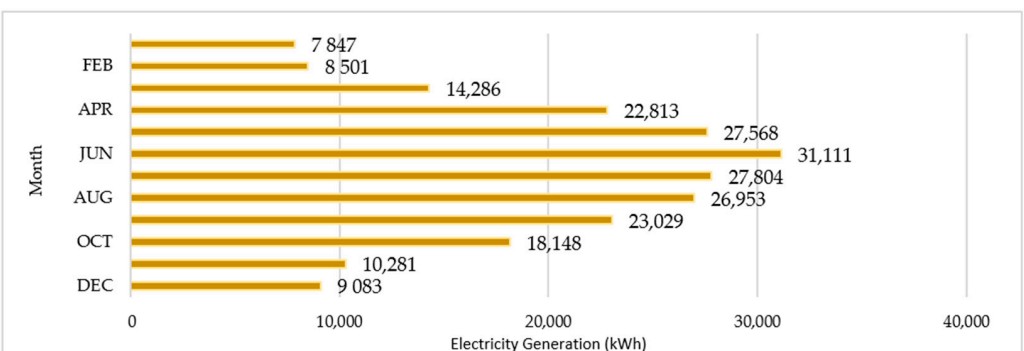

**Figure 2.** Total monthly generation of the REC (4 PV systems).

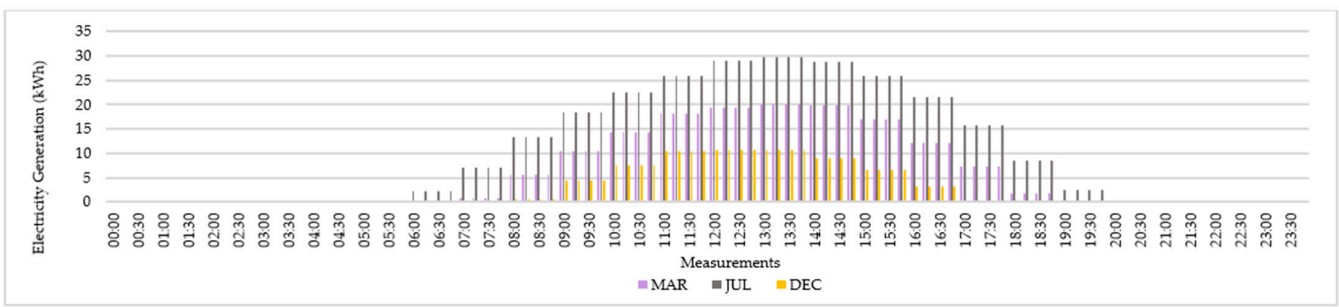

**Figure 3.** Average total generation profile of REC Telheiras on a 15 min basis for an average day of three selected months.

It is important to mention that the months shown in Figure 3 are the months with lower energy generation (December), medium generation (March), and higher energy generation (July).

European countries such as Portugal, Spain, and Italy, especially the southern regions, experience high levels of global horizontal irradiation during the entire year when compared to northern countries such as Belgium, the Netherlands, and the United Kingdom [69]. Summer months or close to this season present the highest values of solar irradiation, justifying the fact that the months with higher PV generation are July, June, August, and May, respectively. Also, as expected, the hourly results for the generated energy match the solar radiation profile, with a peak value close to the solar noon (the moment when the sun reaches the highest position in the sky) [70].

System C (Gymnasium) has a significant importance for the total PV generation of the four buildings in the REC due to its significant roof availability (approximately 68% of the whole generation is produced using System C). Since the acquisition of equipment for PV systems in the case study is performed through collective investment by the members, it might be challenging to install high-capacity systems as System C during the initial phase of the REC, and possible divisions into stages at different times could be a valuable solution for this specific PV system.

After the simulation of the four analyzed PV systems and with the information provided in the methods section, the total costs of each system and the global costs were calculated. Thereby, System A will require EUR 6377 of investment; System B will need EUR 17,978; System C EUR 48,979 of investment; and EUR 28,747 for System D, totalizing a global cost of EUR 97,220. Appendix A presents a detailed view of all the unitary costs and equipment required for each PV system.

### 3.2. Consumption Profile Analysis

#### 3.2.1. Average Household Member Consumption Profile

The monthly average consumption values for the average member are shown in Figure 4 and the member consumption profile obtained through the average of eight households that are members of REC Telheiras is presented in Figure 5. As mentioned, this analysis was based on 15 min periods and average consumption days for each month.

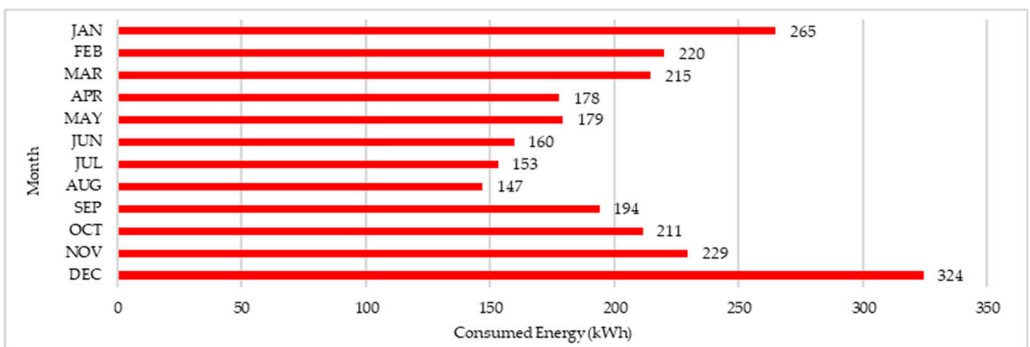

**Figure 4.** Total monthly consumption—average member.

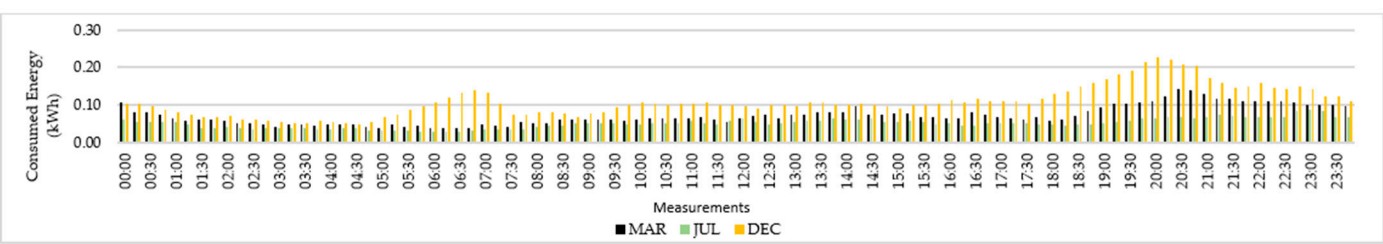

**Figure 5.** Average consumption profile of a household member of REC Telheiras on a 15 min basis for an average day of three selected months.

Analogous to what was undertaken for energy generation values in Figure 4, not all the months are presented in Figure 6 (only March, July, and December).

It is noticeable that the months with higher energy consumption are winter months (lower average temperatures where heating systems are more demanded and possibly supplied using electrical equipment), namely December, January, November, and February, possibly related to electric heating systems promoting a higher level of electricity consumption during colder days [71]. The opposite occurs during the summer; the lowest energy consumption levels are observed in August, July, and June, respectively. Also, the obtained average profile is in accordance with the pattern observed for household electricity consumption, facing peaks during the early morning and the evening [11].

Here, it is also important to mention that a household energy consumption profile varies considerably according to specific behaviors associated with internal and external aspects, such as environmental awareness, energy consumption patterns, income, housing characteristics, and number of residents [72]. Household energy use can also be linked to the health conditions of its residents through ambient climatization and quality of indoor air [73], age of the residents [74], and interior thermal comfort [75,76].

The obtained values match the expected for a household; the annual average energy consumption obtained from the analysis of the profiles of all eight members of REC Telheiras is 2474.87 kWh/year, close to the average household consumption of a Portuguese household (2384.8 kWh/year in 2022) [77].

### 3.2.2. Public Buildings Electricity Consumption Profiles

This subsection presents the consumption profiles for the four public buildings whose rooftops were used for the PV system simulation. Since this community center is not used daily and is more associated with periodic local events, courses, and meetings, relatively low average consumption levels are observed during a typical day, with considerable peaks during the morning and afternoon of winter days.

It is important to mention that the obtained consumption profile matches the opening hours of the building (from 9:00 a.m. to 1:00 p.m. and from 2:00 p.m. to 5:00 p.m.). This justifies the considerably lower consumption at night and in the early morning, as well as a significant decrease in the measured consumption between 1 p.m. and 2 p.m. Particularly during the typical months of working and academic holidays in Portugal (a fraction of July and the entirety of August), the lowest consumption values are obtained for this building, as a significantly smaller number of local events occurs since many local residents are on vacation.

As local government entities are responsible for many aspects of their territory, such as the management, maintenance, and cleaning of public spaces and equipment, housing and community intervention, and local recreational and cultural activities, a considerable number of people work inside its headquarters (Building B), where climatization, office, and data management equipment are required. In this specific case, the headquarters is a historical building, potentially with a low energy performance and a relatively large area and many different office spaces and meeting rooms.

Considerable high values of energy consumption are observed during the opening hours of Building B (from 9:00 a.m. to 5:00 p.m.), facing peaks during winter days possibly due to the usage of heating systems. In addition, slightly higher values during the morning may be associated with a larger movement of citizens attending services during these hours.

Building C is dedicated to the sport practices of local clubs and is open to citizen use upon payment of an hourly fee during the entire week. Since most citizens do not have available time in the morning and afternoon due to work, school, or other regular occupations, the highest levels of energy consumption of Building C are observed during the evening, particularly from 8:00 p.m. to 11:00 p.m., which can be associated with the regular practice of collective sports and activities. It is also noticeable that the monthly consumption values vary strongly, which may also be linked to the courses schedule and variations in the number of personal reservations.

The primary school (Building D) is centrally located in the neighborhood, encompassing students in the first basic cycle and normally between 6 and 9 years of age, with different classrooms, a cafeteria, and two outdoor sports courts. The school was recently renovated, and air conditioning systems were placed in classrooms.

The obtained average consumption profiles of each public building for an average day of selected months and the total monthly consumption are presented in Figure 6a–h.

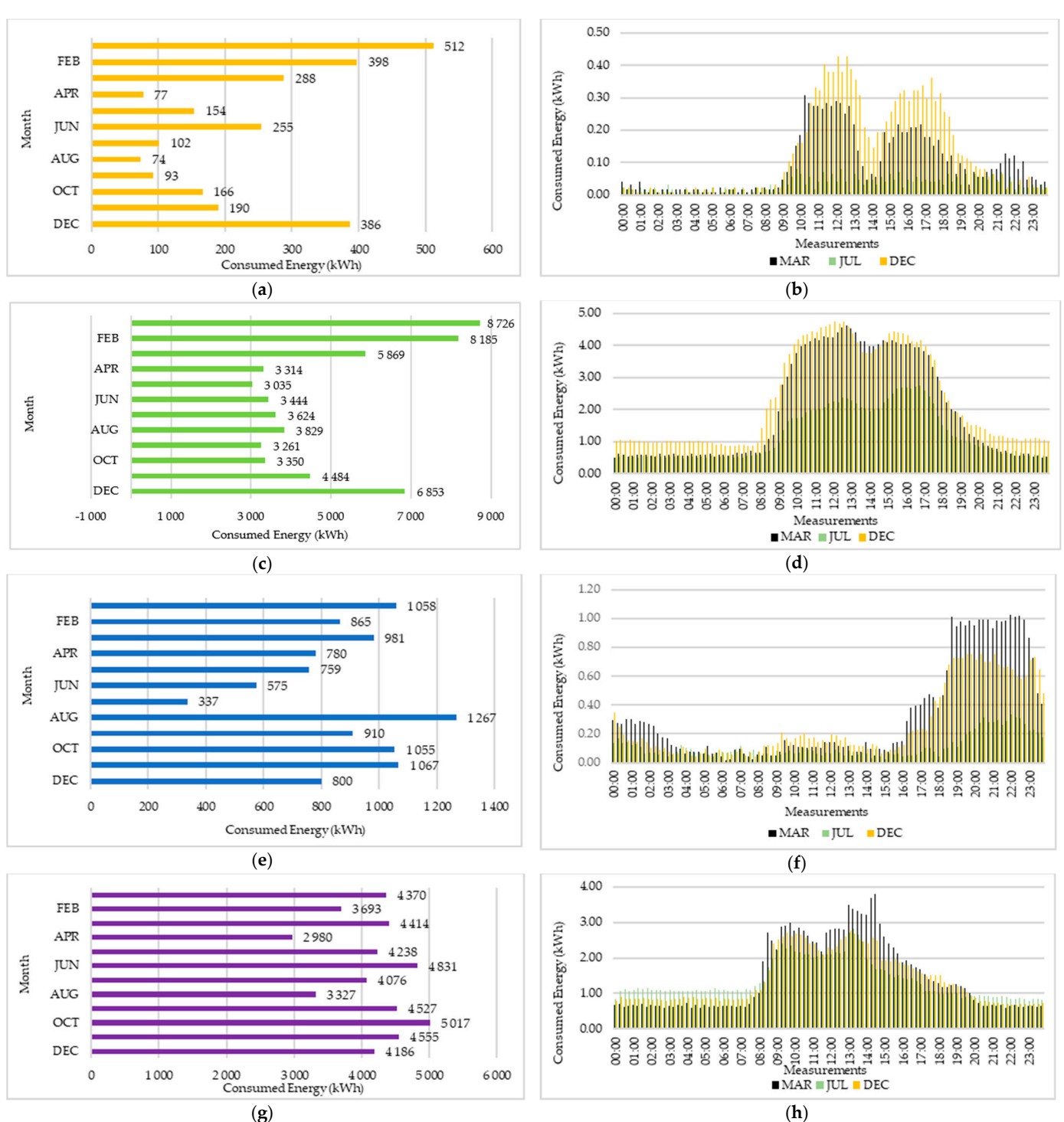

**Figure 6.** Average daily consumption profiles for typical days and monthly total consumption: (**a**,**b**) Building A; (**c**,**d**) Building B; (**e**,**f**) Building C; (**g**,**h**) Building D.

### 3.3. Analysis of Energy Compensation: SCI and SSI Indexes

3.3.1. Scenario A: Current Portuguese Legislation for Energy Sharing

In this scenario, the generated energy must be self-consumed locally in the generation site before being shared with other REC members as much as possible. For each of the four public buildings, the energy compensation was calculated, and the annual average values of SCI and SSI indexes can be seen in Figure 7. The average daily surplus for three

selected months and the monthly total surplus for each public building are presented in Figure 8a–d.

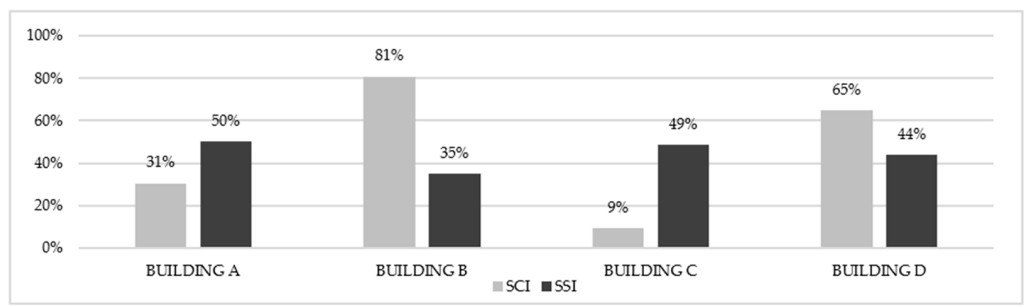

**Figure 7.** Scenario A: average annual SCI and SSI for each analyzed public building.

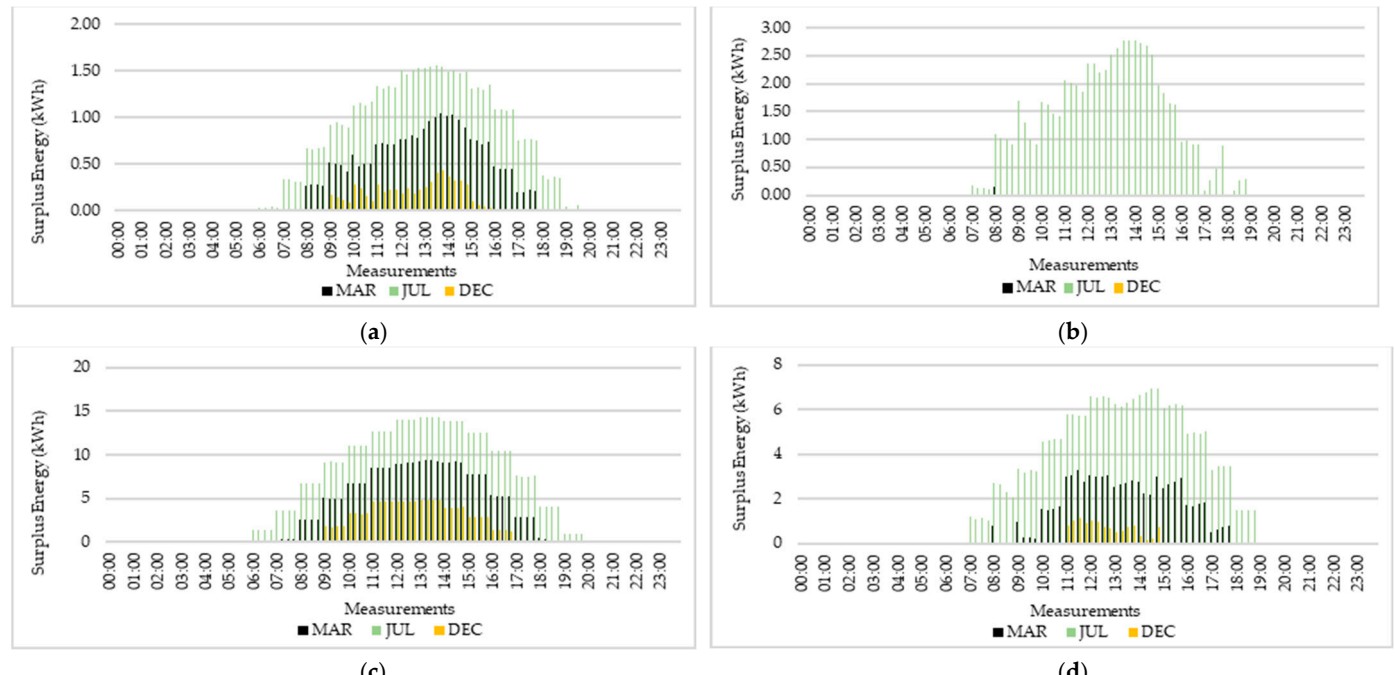

**Figure 8.** Average surplus energy profile on a 15 min basis for an average day of three selected months: (**a**) Building A; (**b**) Building B; (**c**) Building C; (**d**) Building D.

Analyzing the results, it becomes evident that the public buildings with higher energy consumption, particularly during hours of sunshine (Buildings B and D), present higher values for SCI and lower values for SSI (since higher consumption values during the day occur, a larger amount of the generated energy will be locally self-consumed, as well as a lower percentage of the total consumption will be supplied through distributed generation). Thereby, the opposite occurs in Buildings A and C, where lower levels of SCI and higher levels of SSI are reported.

It is also important to emphasize that results depend on the generation of the respective PV system. In the case of Building C, the generation represents almost 50% of the total annual generation of the REC (48%), which also contributes to a considerably lower value of SCI = 9% (total generation is significantly larger than the self-consumption fraction) and a substantially higher value of SSI = 49% (during the daytime, in most of the 15 min periods, almost half of the consumption will be supplied through local generation).

Figure 9 brings a view of the total surplus energy of each analyzed public building in each month of the year.

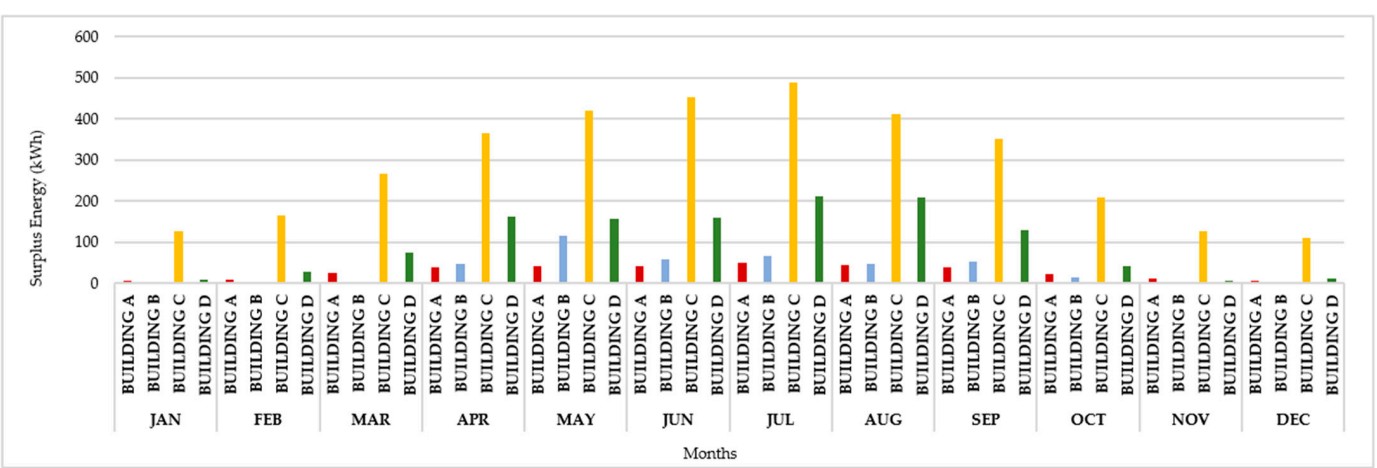

**Figure 9.** Monthly surplus energy according to each building and for each month of the year.

Evaluating the results shown in Figure 9, the months with higher surplus energy are summer months—particularly July—where the average lower energy consumption levels are observed. This highlights the fact that these months have the lowest levels of SCI and the highest levels of SSI. Also, it is important to emphasize that the daily peak of surplus energy of all the analyzed buildings is close to midday, following the PV generation profile and the peak of solar irradiance.

Then, the total surplus energy from the four public buildings, for each of the 15 min periods of a typical day for each month, was used to calculate the SCI and SSI indexes for the residential members of the REC according to different numbers of participants. After multiple interactions, ten different numbers of members were selected for Scenario A, namely 10, 30, 50, 70, 90, 110, 130, 150, 170, and 190 members, where the results are presented in Figure 10.

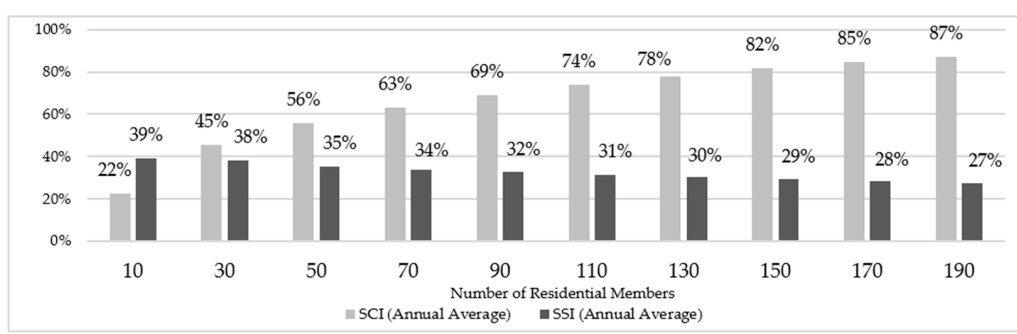

**Figure 10.** SCI and SSI for the residential members according to the number of participants in the REC for Scenario A.

Analyzing the results, it is possible to conclude that increasing the number of participants in the REC results in an increase in SCI and a decrease in SSI. This occurs since the rise in the number of members promotes an increase in consumption while the surplus energy available to share among members stays the same, resulting in lower self-consumption values and a higher total consumption of the REC.

It is also important to note that relatively high SSI values will not be obtained, even for low numbers of members, as a significant part of the average consumption for a member of REC Telheiras occurs during the evening (the lowest possible value of SSI, for only one residential member, is 40%). On the other hand, the maximum value of SCI (100%) is obtained with at least 491 residential members, with an associated value of SSI = 15%.

In this scenario, the number of residential members that maximizes both SCI and SSI is only 22 members (SCI = 38% = SSI); this extremely low number can be explained by the

high level of household energy consumption during the night, where no PV generation occurs. This quantity of members is deeply unfavorable for the REC since there will be an enormous amount of surplus energy being injected into the grid and sold at lower tariffs, a few members will be responsible for a considerable investment in all four PV systems, and the associated payback time will be considerably higher than in other possible scenarios.

### 3.3.2. Scenario B: Fixed Sharing Coefficient for All Members including Public Buildings

In this scenario, all the generated energy is shared under a fixed and equal coefficient among the members and the four analyzed public buildings. Since the main objectives of the REC are energy sharing between citizens of the neighborhood and the local mitigation of energy poverty, not maximizing energy self-consumption in public buildings, the evaluation of SCI and SSI was performed to maximize the indexes for the residential members.

After multiple interactions, ten different numbers of residential members were selected for Scenario B: 10, 50, 90, 130, 170, 210, 250, 290, 330, and 370 members. The results can be seen in Figure 11.

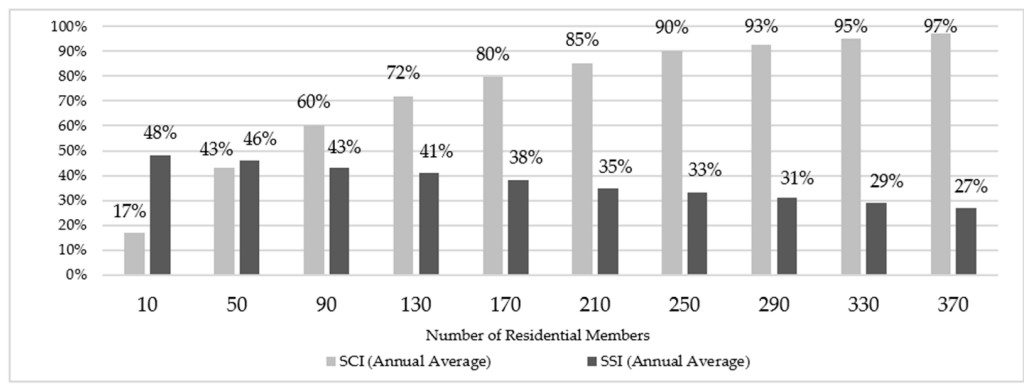

**Figure 11.** SCI and SSI for residential members according to the number of participants in the REC for Scenario B.

For Scenario B, it is important to mention that since the energy is equally shared between the members and the public buildings, the analysis according to a specific number of participants in the REC means that energy is being shared between this number of residential members plus four other members, corresponding to the four analyzed buildings.

The same observed pattern of variation for SCI and SSI for Scenario A occurs in Scenario B; SCI increases and SSI decreases with the increase in the number of residential participants. However, given that all generated energy is available to share between the members, more participants are required to achieve higher values of SCI and, consequently, to obtain the number of members that maximizes both values for SCI and SSI; in this context, this can be obtained with 55 members (SCI = 45% = SSI). In the same situation as Scenario A, this also represents an extremely low number of members and is really unfavorable for the REC development. In addition, the lowest value of SSI is obtained for only one residential member (SSI = 49% and SCI = 9%), as well as SCI = 100% being obtained with at least 583 participants in the REC, with a respective value of SSI = 19%.

As aforementioned, the average annual values of SCI and SSI for each public building and with 153 participants were calculated (Figure 12). It becomes clear that an extremely lower value of SSI occurs for buildings with higher consumption levels, especially during the day (Buildings B and D), as well as a higher SCI value for buildings with lower consumption levels (Buildings A and C).

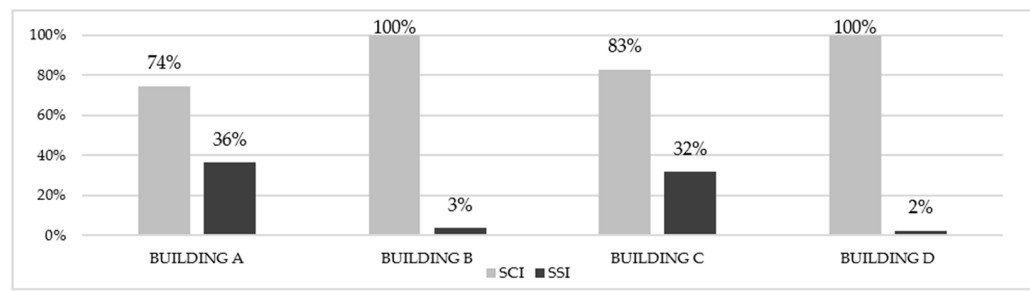

**Figure 12.** Scenario B: SCI and SSI average annual values for each public building and considering 153 members.

*3.4. Scenarios Comparison*

Evaluating both scenarios, it becomes clear that the most important difference between them is the number of members; with the same number of participants, higher values of SCI for the members are observed in Scenario A than in Scenario B, as well as lower values of SSI since higher values of generated energy are available in Scenario B. Figure 13 brings a view of how many residential members are needed to obtain certain values of SCI in each scenario. Figure 14 presents the associated SSI values.

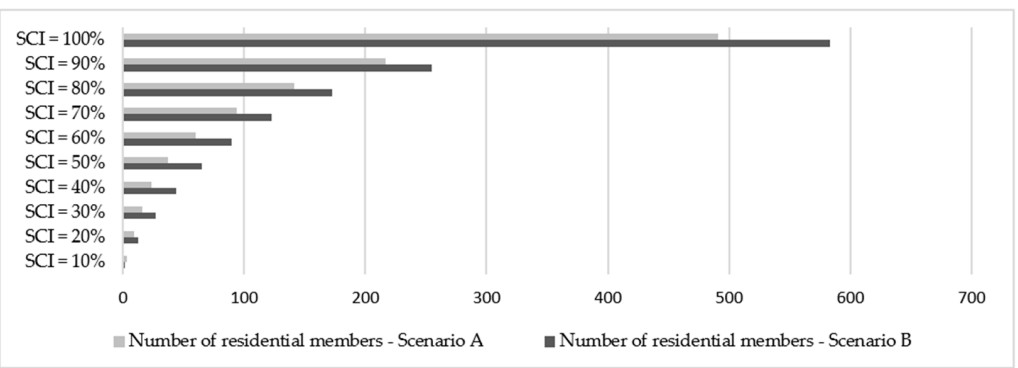

**Figure 13.** Number of residential members required to obtain specific values of SCI in each scenario.

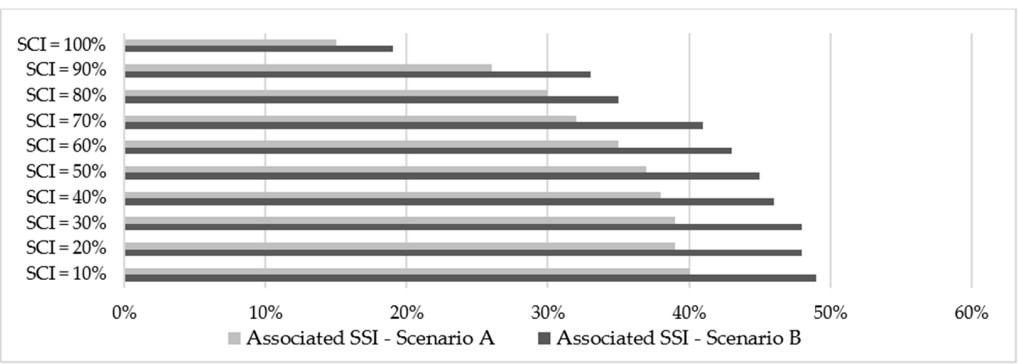

**Figure 14.** Specific values of SCI and its associated SSI for each scenario.

Through analyzing both figures, it is possible to conclude that, for the same value of SCI, not only will the number of residential members be higher for Scenario B, but higher associated values of SSI will also occur, highlighting the fact that this scenario is more interesting for the residential members according to both indexes.

Two scenarios can possibly occur inside a REC with the aim to define its optimized number of members: (a) maximizing both SCI and SSI equally, aiming to provide a community that injects into the grid the least and is energetically self-sufficient as possible;

or (b) providing 100% or a value as high as possible of SCI, aiming to avoid injection into the grid as much as possible. In both cases, Scenario B achieves the objective with a higher number of members, symbolizing that more families could experience the economic and social benefits of being part of REC Telheiras with the same PV systems in the same public buildings.

The results of the financial comparative analysis are presented in Tables 7 and 8. Both scenarios were analyzed according to the lowest number of families that provide SCI = 100% (419 families for Scenario A, 583 families for Scenario B).

**Table 7.** Financial analysis—Scenario A.

| REC Members | | Investment (EUR) | Available Energy (kWh/Year) | Annual Grid Costs (EUR/Year) | Annual O and M Costs (EUR/Year) | Energy Bills Savings (EUR/Year) |
|---|---|---|---|---|---|---|
| Local government | A | 375 | 835 | - | 11.25 | 134 |
| | B | 21,078 | 46,959 | - | 632 | 7513 |
| | C | 4223 | 9409 | - | 127 | 1505 |
| | D | 14,650 | 32,639 | - | 440 | 5222 |
| | Total | 40,326 | 89,842 | - | 1210 | 14,374 |
| Families | 1 family | 147.39 | 328 | 3.48 | 4.19 | 52.54 |
| | Total | | | | | |
| | ($n$ = 419) | 61,756 | 137,432 | 1458 | 1852 | 22,014 |

**Table 8.** Financial analysis—Scenario B.

| REC Members | | Investment (EUR) | Available Energy (kWh/Year) | Annual Grid Costs (EUR/Year) | Annual O and M Costs (EUR/Year) | Energy Bills Savings (EUR/Year) |
|---|---|---|---|---|---|---|
| Local government | A | 174 | 387 | - | 5.22 | 62 |
| | B | 174 | 387 | - | 5.22 | 62 |
| | C | 174 | 387 | - | 5.22 | 62 |
| | D | 174 | 387 | - | 5.22 | 62 |
| | Total | 696 | 1548 | - | 20.88 | 248 |
| Families | 1 family | 174 | 387 | 4.11 | 5.22 | 62 |
| | Total | | | | | |
| | ($n$ = 583) | 101,442 | 225,621 | 2396 | 3043 | 36,146 |

Important aspects are presented in this financial analysis: In both cases, the payback time of a residential member (3.3 years) will be slightly higher than the one for a public building (3.1 years), since only the families will need to pay grid access fees. In addition, it becomes clear that Scenario B provides more annual energy for each family than Scenario A (more 59 kWh/year), resulting in more energy bill savings. However, since the amounts of generated energy for public buildings and families in Scenario A are different, a residential member needs to make a lower investment to be part of the community (EUR 27 less than in Scenario B) and needs to pay a lower value for operation and maintenance. With relatively short payback times on investment, this analysis showcases the potential of RECs to share the economic benefits of renewable energy generation between local governments and hundreds of local families.

Since one of the main objectives of a REC is to promote local development to its members in many different areas (economic, social, and environmental) and to mitigate energy poverty, it should aim to include a greater number of citizens and share the benefits from the participation in a REC. This is achieved for Scenario B with the possibility of using fixed coefficients to share the full generation potential inside the REC. While Scenario B can maximize the number of local households that are able to join the REC, it also reduces the benefits for the local government. In the future, a mixed approach could also be pursued with different fixed, proportional, and/or dynamic sharing coefficients for the members according to their consumption, investment levels, and/or other factors.

## 4. Conclusions

Renewable energy communities can play a relevant role in the success of the current energy transition, contributing to a multidimensional process of transforming citizens from energy consumers into active players [78]. Case studies, such as the REC Telheiras explored herein, illustrate the concept's applicability, and may pave the way for other neighborhoods, cities, and regions to produce their own energy communities. In the context of low incomes, rising energy costs, and low energy performance of buildings, many European Union [79] and Portuguese [80] citizens are in energy poverty. Renewable energy communities can partially address this problem through the participation of energy-poor households in energy-sharing schemes and through other local energy poverty mitigation activities. In this context, analyzing the match between distributed generation systems—such as PV systems—and household electricity consumption according to the current legislation for energy sharing in a REC is crucial for determining the possible optimized number of members and evaluating the investment required per member in possible expansions.

The implementation process of REC Telheiras and its associated challenges and solutions can serve as a guideline for other energy citizenship initiatives. Independently of the country, legislation and scenario, the challenges faced in the creation of REC Telheiras may be the same for all citizen-led energy organizations, where the solutions applied to this study case may be replicated for other organization and geographies.

In the case study, all analyzed PV systems were sized for roofs of public buildings, evidencing the importance of partnerships between RECs and local authorities. However, installing PV panels on private buildings' roofs is possible but may translate into a bigger complexity for the community's organizational structure. These may include administrative and legal questions regarding the utilization of private roof areas and the ownership of the equipment, the necessity of revision of the internal regulations of the REC, and the necessity of maintaining the main objectives of the REC (local development of the community, not financial profits).

The results obtained from the PV systems highlight the importance of selecting buildings with large available roof areas, few shading effects, and a satisfactory overall rooftop condition. In the case study, most PV generation will come from Buildings C and D, which account for 48% and 30% of the total annual generation, respectively. Since REC Telheiras is still in a pilot phase with Building A, two possible scenarios would be appropriate to the current reality: (a) installing System B as the second system of the REC because it will have a lower cost and, consequently, will require a lower number of new members; or (b) installing fractions of Systems B or C, where future expansions can be performed according to the recruitment of new members.

The differences in the obtained results of SCI and SSI for Scenario A and Scenario B show how the current Portuguese legislation hinders the participation of more members in RECs by mandating that the generated energy must be locally self-consumed by the installation site itself and only the surplus energy is available to share among the other members. This requirement reduces the number of participants for the same PV generation values to achieve conditions of high values of SCI or even no injection into the grid (100% of SCI). Since smart meters are already widely applied in Portuguese buildings—either households, public buildings, or commercial sites—the analyzed possibility of Scenario

B should, in theory, be applicable, which could be an incentive for the emergence of more initiatives like REC Telheiras all over the country. Greater flexibility for RECs to choose their energy-sharing approach could further increase the concept's attractiveness, enhance the number of RECs in Portugal, and accelerate licensing processes. These changes would better align policy expectations towards carbon neutrality and renewable energy integration with on-the-ground real-world case developments and barriers. It is important that the current legislation regarding RECs in Portugal aligns with the real challenges and difficulties in implementing these projects, focusing on questions related to methods of sharing the generated energy, awareness raising and capacity building, funding schemes to foster the creation of new RECs, and more agile licensing processes.

In this context, it is important to highlight the importance of smart metering since this enables energy-sharing schemes, as well as the characterization and identification of household consumption patterns and possibilities to apply different energy efficiency and flexibility measures [81]. In Portugal, there are over 5.8 million smart meters installed, where most Portuguese households can obtain access to instant data regarding energy consumption and be part of collective energy schemes [82].

Further research can analyze other possible forms of policies regarding energy compensation, such as net metering schemes or different energy-sharing coefficients associated with the consumption of each member/public building or with the initial investment of the participant. Furthermore, smart meter data can be used to adjust households' energy use to better match PV generation. Renewable energy communities are just emerging in the European Union, and more research supporting empirical case studies is needed if their promise of a more sustainable, democratic, and just energy system is to be fulfilled.

**Author Contributions:** Conceptualization, E.F. and J.P.G.; Methodology, E.F. and M.M.S.; Software, E.F.; Validation, M.M.S.; Investigation, E.F. and M.M.S.; Writing—original draft, E.F.; Writing—review & editing, M.M.S. and J.P.G.; Project administration, M.M.S. and J.P.G. All authors have read and agreed to the published version of the manuscript.

**Funding:** The authors acknowledge and are thankful for the support provided to CENSE by the Portuguese Foundation for Science and Technology (FCT) through the strategic project UIDB/04085/2020. Miguel Sequeira PhD scholarship is funded by Fundação para a Ciência e Tecnologia (2020.04774.BD).

**Institutional Review Board Statement:** Not applicable.

**Informed Consent Statement:** Not applicable.

**Data Availability Statement:** Data are contained within the article.

**Acknowledgments:** The authors thank the members of REC Telheiras for their collaboration.

**Conflicts of Interest:** The authors declare no conflict of interest.

### Appendix A. Detailed Costs of the Four Analyzed PV Systems

**Table A1.** Detailed costs of the four analyzed PV systems.

| PV System | Components | Quantity | EUR |
|---|---|---|---|
| | PV Module Risen RSM 150-8-500-M | 16 | 3840 |
| | Inverter Sungrow SG8.0 RT | 1 | 1143 |
| | Aluminium Profile 2.08 m | 40 | 460 |
| **System A** | End Clamp 30/35 mm | 16 | 15.52 |
| | Middle Clamp 30/35 mm | 24 | 12.24 |
| | Roof Hook Stainless Steel | 56 | 253 |
| | Solar Cable 6 mm$^2$ Black (m) | 70 | 120 |
| | Solar Cable 6 mm$^2$ Red (m) | 70 | 120 |

**Table A1.** *Cont.*

| PV System | Components | Quantity | EUR |
|---|---|---|---|
| **System A** | Protection Cable 6 mm$^2$ Yellow/Green (m) | 70 | 111 |
| | Installation Costs | - | 304 |
| | Total Costs of System A | | 6377 |
| **System B** | PV Module Risen RSM 150-8-500-M | 58 | 13,920 |
| | Inverter Sungrow SG20.0 RT | 1 | 1628 |
| | Aluminium Profile 2.08 m | 48 | 552 |
| | End Clamp 30/35 mm | 16 | 15.52 |
| | Middle Clamp 30/35 mm | 88 | 44.88 |
| | Roof Hook Stainless Steel | 80 | 362 |
| | Solar Cable 6 mm$^2$ Black (m) | 120 | 205 |
| | Solar Cable 6 mm$^2$ Red (m) | 120 | 205 |
| | Protection Cable 6 mm$^2$ Yellow/Green (m) | 120 | 190 |
| | Installation Costs | - | 856 |
| | Total Costs of System B | | 17,978 |
| **System C** | PV Module Risen RSM 150-8-500-M | 156 | 37,440 |
| | Inverter Growatt 70KTL3-X | 1 | 4797 |
| | Aluminium Profile 2.08 m | 140 | 1610 |
| | End Clamp 30/35 mm | 36 | 34.92 |
| | Middle Clamp 30/35 mm | 300 | 153 |
| | Roof Hook Stainless Steel | 246 | 1112 |
| | Solar Cable 6 mm$^2$ Black (m) | 300 | 513 |
| | Solar Cable 6 mm$^2$ Red (m) | 300 | 513 |
| | Protection Cable 6 mm$^2$ Yellow/Green (m) | 300 | 474 |
| | Installation Costs | - | 2332 |
| | Total Costs of System C | | 48,979 |
| **System D** | PV Module Risen RSM 150-8-500-M | 93 | 22,320 |
| | Inverter Sungrow SG25.0 RT | 1 | 1991 |
| | Aluminium Profile 2.08 m | 70 | 805 |
| | End Clamp 30/35 mm | 20 | 19.40 |
| | Middle Clamp 30/35 mm | 180 | 91.80 |
| | Roof Hook Stainless Steel | 144 | 651 |
| | Solar Cable 6 mm$^2$ Black (m) | 300 | 513 |
| | Solar Cable 6 mm$^2$ Red (m) | 300 | 513 |
| | Protection Cable 6 mm$^2$ Yellow/Green (m) | 300 | 474 |
| | Installation Costs | - | 1369 |
| | Total Costs of System D | | 28,747 |
| | Total Costs of All the Four Analyzed PV Systems | | 102,082 |

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
