# Peer review of "Sharing Is Caring: Exploring Distributed Solar Photovoltaics and Local Electricity Consumption through a Renewable Energy Community"

_sustainability, doi:10.3390/su16072777_

Round 1
Reviewer 1 Report
Comments and Suggestions for Authors
Comments and Suggestions for Authors.
Manuscript ID: sustainability-2916743
Type of manuscript: Article
Title: Sharing is Caring: Exploring distributed solar photovoltaics and local electricity consumption through a Renewable Energy Community
Authors
Evandro Cahino Ferreira, Miguel Macias Sequeira, João Pedro Gouveia
Interesting article promoting solar-to-electricity conversions in the local community. For further procedure, please make the following additions:
1. the authors chose a monocrystalline module for installation, justifying this choice very poorly. Please complete section 2.2 with a comparison with other photovoltaic modules, e.g. polycrystalline, amorphous, perovskite.
2. Please complete the description of how to account for excess electricity depending on the type of facility (A,B,C,D - net billing, net metering) in the Portuguese energy law.
3. In the summary, please supplement with the possibility of using research.
4. in the summary, please supplement with information on whether the results of the research can be transferred to other facilities, e.g. industrial, and to what extent.
5. in section 2.1. please provide information on the costs of the various installations and the sources of their financing.
All the above comments should be taken into account in the work.
Author Response
Thank you for the comments on our manuscript and valuable suggestions. Please, see the attachment.

Reviewer 2 Report
Comments and Suggestions for Authors
The article discusses the role of Renewable Energy Communities (RECs) in enhancing citizen participation in the energy transition within the European Union. It focuses on a case study from REC Telheiras, Lisbon, and evaluates Portuguese legislation for energy sharing through two scenarios. As a first note, the manuscript is informative and provides a valuable contribution to understanding RECs in the energy transition. With the following comments including a few suggested improvements, it has the potential to serve as a significant resource for policymakers, researchers, and energy communities alike.
1. Clarity and Context: The report provides a clear definition of RECs and outlines their importance in the energy transition. However, it could benefit from a more detailed explanation of the European Union’s role and its impact on RECs to set a stronger context for readers unfamiliar with the subject.
2. Relevance of RECs: The report successfully underscores the relevance of RECs in the energy transition. It would be beneficial to further emphasize the multidimensional benefits that RECs bring to communities, such as economic, social, and environmental impacts. The analysis of legislative support for RECs is thorough, but the report could delve deeper into the specific provisions that have led to the slow uptake of RECs. A comparative analysis with other Member States could provide valuable insights into the challenges faced.
3. Case Study Evaluation: The selection of the REC Telheiras, Lisbon, as a case study is appropriate and well-justified. The report effectively describes the photovoltaic systems and their capacity. However, it would be beneficial to include more information on the community’s demographic and socioeconomic profile to understand the energy-sharing dynamics better. While the REC Telheiras case study is well-presented as an example of REC applicability, the article could expand on how these insights can be generalized or adapted to other contexts, considering the unique challenges and opportunities in different regions.
4. Methodological Rigor: The methodology for matching local generation with household consumption is sound. Nonetheless, the report should address the two scenarios' scalability and applicability to other RECs. The distinction between PV systems in public versus private buildings raises important points about complexity in organizational structure. The article could elaborate more on why “installing PV panels on private buildings' roofs may translate into a bigger complexity for the community's organizational structure” (ref. 2ns para. p.18). It could suggest potential strategies for managing this complexity and fostering successful partnerships.
5. Data and Analysis: The data presentation is clear, and the analysis is logical. The article successfully interprets the simulation results but could enhance its impact by including visual aids such as charts or graphs to illustrate key findings. The discussion on energy poverty is significant and timely. The report could delve deeper into how RECs specifically address energy poverty, perhaps by providing examples of successful energy-sharing schemes that have alleviated such conditions. The comparison of scenarios A and B is insightful. The report could benefit from a more detailed discussion on the cost-benefit analysis of each scenario, considering not only the number of new members required but also the long-term sustainability and energy independence of the REC.
6. Implications and Conclusions: Energy policy and community engagement implications are well-drawn. The discussion on energy poverty is significant and timely. The report could delve deeper into how RECs specifically address energy poverty, perhaps by providing examples of successful energy-sharing schemes that have alleviated such conditions.
7. Conclusions: The conclusions are robust and align with the objectives of the manuscript. To add value, the article could suggest practical steps for RECs to implement the recommended changes. The article's conclusion effectively highlights the potential of Renewable Energy Communities (RECs) in transforming the energy landscape. The call for greater flexibility in energy-sharing approaches is powerful. The report could suggest ways to better align policy with the practical realities of REC operation and expansion. While the article concludes on a strong note, it could further discuss the broader implications of its findings for future policy development and the scaling up of RECs.
Overall, the manuscript provides a compelling argument for the role of RECs in the energy transition. The article rightly emphasizes the importance of selecting suitable buildings for PV installations. It could be enhanced by discussing potential criteria or decision-making tools that could aid in the selection process for future projects. For example, the mention of smart meters as an enabler for Scenario B is a key point. The article could explore how existing smart meter infrastructure can be leveraged to support more innovative and flexible energy-sharing models. The suggested enhancements can offer a more comprehensive view of the challenges and opportunities associated with RECs, thereby contributing valuable insights for policymakers, practitioners, and researchers in the field.
Comments on the Quality of English LanguageThe language quality is clear and sound. Some very minor typos that a quick spell-check could resolve are present, but one item might need a second look: the article should ensure that all technical terms and acronyms are clearly defined at their first instance (e.g., SSI, SCI, etc.) for the benefit of all readers, including those who may not be familiar with the field since this article would most likely be interesting for people from various fields and backgrounds.
Author Response

(The authors gave the same response as above.)
